# The Porous Structure of Peripheral Nerve Guidance Conduits: Features, Fabrication, and Implications for Peripheral Nerve Regeneration

**DOI:** 10.3390/ijms241814132

**Published:** 2023-09-15

**Authors:** Teng Wan, Yi-Lin Wang, Feng-Shi Zhang, Xiao-Meng Zhang, Yi-Chong Zhang, Hao-Ran Jiang, Meng Zhang, Pei-Xun Zhang

**Affiliations:** 1Department of OrthopedSics and Trauma, Peking University People’s Hospital, Beijing 100044, China; tengwan.med@hotmail.com (T.W.);; 2Key Laboratory of Trauma and Neural Regeneration, Peking University, Beijing 100044, China; 3National Centre for Trauma Medicine, Beijing 100044, China

**Keywords:** porous structure, peripheral nerve guidance conduit, peripheral nerve regeneration, permeability

## Abstract

Porous structure is an important three-dimensional morphological feature of the peripheral nerve guidance conduit (NGC), which permits the infiltration of cells, nutrients, and molecular signals and the discharge of metabolic waste. Porous structures with precisely customized pore sizes, porosities, and connectivities are being used to construct fully permeable, semi-permeable, and asymmetric peripheral NGCs for the replacement of traditional nerve autografts in the treatment of long-segment peripheral nerve injury. In this review, the features of porous structures and the classification of NGCs based on these characteristics are discussed. Common methods for constructing 3D porous NGCs in current research are described, as well as the pore characteristics and the parameters used to tune the pores. The effects of the porous structure on the physical properties of NGCs, including biodegradation, mechanical performance, and permeability, were analyzed. Pore structure affects the biological behavior of Schwann cells, macrophages, fibroblasts, and vascular endothelial cells during peripheral nerve regeneration. The construction of ideal porous structures is a significant advancement in the regeneration of peripheral nerve tissue engineering materials. The purpose of this review is to generalize, summarize, and analyze methods for the preparation of porous NGCs and their biological functions in promoting peripheral nerve regeneration to guide the development of medical nerve repair materials.

## 1. Introduction

The application of tissue-engineered conduits in peripheral nerve regeneration has been extensively studied and successfully implemented in clinical practice [1]. Although the peripheral nervous system (PNS) possesses inherent regenerative capabilities, simple transverse or small-scale injuries can be effectively treated using epineural and/or fascicular sutures [2]. However, larger gaps in nerve segments (>3 mm) present a challenge, as axonal reconnection cannot occur. Autologous nerve grafting (typically from the sural nerve of the patient) has become the “gold standard” for treating long-gap nerve defects. Autologous cells provide a natural nerve structure that serves as a template for regeneration. Additionally, extracellular matrix proteins (ECMs), Schwann cells, and growth factors create an ideal microenvironment for peripheral nerve regeneration [3]. However, autologous grafts also have inherent disadvantages, including limited donors, loss of donor function, neuroma formation, nerve distortion or dislocation, and nerve diameter mismatch. Tissue reperfusion following autologous nerve transplantation induces apoptosis or necrosis. These limitations of autografts have prompted the development of alternative therapeutic options to improve patient outcomes [4].

In this context, tissue-engineered peripheral nerve guidance conduits (NGCs) have emerged as a viable solution to overcome the limitations of autologous tissues [5]. Typically, NGCs are cylindrical tubular structures composed of degradable or nondegradable synthetic or natural materials that are implanted at the site of a nerve defect to facilitate specific regeneration of proximal axons towards their distal target organ [6]. Numerous artificial peripheral nerve repair grafts based on commercial products have been successfully implemented in clinical practice, including the Avance Nerve Graft (AxoGen, Inc., Alachua, FL, USA), NeuraWrapTM (Integra Life Sciences, Princeton, NI, USA), and NeuroFlex (Collagen Matrix, Princeton, NI, USA). With the development of neural tissue engineering and regenerative medicine, several ideal design requirements have emerged as guiding principles for continued nerve graft development [5]. NGCs in clinical application should have the following properties: (1) excellent bioactivity that can guide axons to grow from the proximal end into the distal stump, avoiding the formation of neuroma [7]; (2) mechanical performance to avoid wall breakage during suture and sufficient flexibility to avoid compression of neural tissue; (3) biocompatibility and nontoxicity to cells and tissue, with almost no immune response [8]; (4) a controllable degradation rate that matches the regeneration phase time of peripheral nerves, avoiding a second operation to remove undegraded implant materials and compression of the regenerated peripheral nerve by undegraded NGCs [9]; (5) a suitable porous structure that can limit the invasion of fibrous scarring into NGCs, hinder axon growth, and facilitate communication of biomolecule signals inside and outside the conduits and between cells and axons [10].

A porous structure of suitable pore size and connectivity was shown to be a determining factor for completion of the biological functions of NGCs, promoting the early adhesion, spreading, proliferation, and differentiation of Schwann cells to form cord-like structures (Bungner bands), promote vascularization, and reduce the formation of fibrous scars [11]. The concept of a nonpermeable silicone “nerve regeneration chamber” presented by Lundborg et al. [12] in the 1990s must be innovated, although its closed space can enrich bioactive factors (such as PDGF, FGF, and TGF-β secreted by Schwann cells or macrophages) and effectively prevent the invasion of fibrous scars. In vivo studies [13,14] have shown that the peripheral nerve repair effects of hollow impermeable conduits are outmatched by those of porous, permeable conduits. The 3D topology of the NGC directly affects the behavior of cells during nerve regeneration. The porosity and permeability of NGCs play important roles in the flow of oxygen, nutrients, and bioactive molecules between the internal and external environments [15]. Thus, porous structures with specific biological functions are critical for positive results in peripheral nerve repair. However, porous structures also have drawbacks; an excessively large pore size causes fibroblast deposition and hinders axon growth [2]. A lack of porosity affects the exchange efficiency of the internal and external walls of the conduits [11]. What type of porous structure should ideal NGCs have? What effect does the pore structure have on the physiological process of peripheral nerve regeneration? These questions do not seem to have clear answers from previous research. The transformation of peripheral nerve tissue repair materials in clinical applications can be promoted through a summary and analysis of the effects of porous structures on the regeneration and repair of NGCs.

As shown in Figure 1, this review reports the latest progress and applications of porous structures in peripheral-nerve-tissue-engineered conduits. The classification of NGCs according to pore structure characteristics and current methods of porosity measurement are summarized. We describe the techniques and principles used in constructing porous structures in NCGs. The physical properties of the porous structure, including the biodegradability, mechanical performance, and permeability of NGCs, in addition to the biological behavior of cells related to peripheral nerve regeneration, are discussed.

## 2. Definition and Measurement of Pore Structure

The porous structure of an NGC refers to the three-dimensional morphology of closed or penetrating voids in biomaterials [16]. The pore structure is characterized by size, connectivity, uniformity, and three-dimensional morphology, which affect the biocompatibility, permeability, density, and mechanical performance of the NGC [11]. Generally, biomaterial pores can be categorized according to their size as macropores (100–500 μm), micropores (<100 μm), and nanopores (<1000 nm) [17]. The pore size of NGCs determines which molecules can be exchanged between the regenerated peripheral nerve and the surrounding microenvironment via the conduit walls. Depending on whether the cells can freely infiltrate the inside of the conduits, NGCs can be categorized as semi-permeable (<10 μm) or fully permeable (>50 μm) membranes [18]. Spherical, tubular, and irregularly shaped porous NGCs have been fabricated; however, with the development of new technologies, more complex structures with higher resolution will also be developed to meet the needs of peripheral nerve regeneration [19]. Pore morphology has attracted more and more attention in NGC design due to its significant influence on both the physiochemical and biological performances of the nerve graft. However, the pore morphology of porous conduits fabricated using traditional methods is usually random, and achieving special customization is difficult. Recently, with the development of lithography processes, electrospinning, and 3D printing technology, complex customized pore morphologies have been constructed. Chakkravarthy et al. [20] prepared a porous Ti-30Nb-2Zr biomimetic scaffold using 3D printing technology. In vitro cytological studies have shown that the pore structure can significantly promote cell adhesion, migration, and protein adsorption of the scaffold, possessing strong clinical application conversion ability. Based on these interconnections, pore structures can be divided into isolated and open structures. Almost all studies have considered open and penetrating pore structures; isolated and closed pore structures fail to achieve material exchange and reduce the mechanical performance of NGCs.

The pore size, porosity, permeability, and connectivity are important in characterizing the porous structure of NGCs, particularly in the development of innovative 3D topographies, as they play a decisive role in nerve regeneration. Conduits constructed from spun fibers are usually characterized by the fiber diameter and thickness. The morphology of porous NGCs can be observed through field emission scanning electron microscopy (SEM) and transmission electron microscopy (TEM); the pore size distribution can be statistically analyzed using image analysis software [16]. This method has been widely used to characterize porous biomaterials. Although it allows for intuitive observation of the 2D morphology of NGCs, it falls short of revealing the complete 3D morphology, and its objectivity is dependent on the observation site.

Porosity is defined as the void volume as a percentage of the total NGC volume [21]. The liquid displacement method in an ethanol medium has been widely used in recent studies. The main steps are described as follows [22]. The conduits are immersed in a known volume (V_1_) of anhydrous ethanol. Ethanol is forced into the pores of the conduit through a series of vacuum-release cycles, and the volume of the solution is recorded as V_2_. The conduits are removed, and the volume is recorded as V_3_. The NGC porosity is calculated using the following equation: porosity rate (%) = (V_1_ − V_3_)/(V_2_ − V_3_) × 100%. This method is simple and convenient, and it is not limited by the experimental equipment; however, the accuracy varies due to the volatility of ethanol, and closed pores cannot be measured. The gas adsorption method is an alternative for measuring the porosity and pore distribution of a conduit and is based on the adsorption of gas molecules on a solid surface [23]. The principle of this method is the exposure of a solid material to a gas environment at pressure, calculating the porosity and pore size distribution of the solid material by measuring the change in gas adsorption [24]. Both approaches have advantages and disadvantages. The liquid displacement method can measure larger pores but is insensitive to pores below the surface tension. The gas adsorption method is sensitive in measuring micropores but ignores the number of macropores and macroscale pores.

Permeability is primarily determined by the size and connectivity of the pore structure. NGCs with larger pores and high connectivity allow molecules and cells to in-fold and out-fold the tube walls more freely [11]. Waste products, such as cellular phospholipid debris due to Wallerian degeneration after peripheral nerve injury, should be released from conduits. Macrophages, vascular endothelial cells, fibroblasts, and Schwann cells gather at the site of peripheral nerve injury and participate in the regeneration of peripheral nerves by releasing bioactive signals via autocrine or paracrine pathways. Thus, the efficiency and mode of substance exchange inside and outside NGCs are important standards for evaluating the conduit repair function. Solutes with different molecular weights, including glucose (Mw 180 Da), lysozyme (Mw 14,600 Da), and BSA (Mw 62,000 Da), were used to simulate the osmotic flow dynamics of signal molecules with different molecular weights in the tube wall [25].

## 3. Fabrication Strategies

For tissue-engineered regenerative materials such as bone tissue repair scaffolds, skin regeneration accessories, and corneal-tissue-engineered materials, methods have been developed for the preparation and construction of porous biomaterials [26]. Generally, these methods are either subtractive or additive to the main material of the nerve repair conduits [27]. Although the preparation strategy for the porous structure of peripheral nerve repair materials is not unique, the specific preparation process has differences owing to the biological functions. For example, a porous structure is constructed in a bone repair scaffold; the main consideration is that a suitable pore structure can provide a point for the early attachment of cells and further promote the functionalization of pre-osteoblasts. The porous structure of the graft plays a key role in the biological responses of Schwann cells, macrophages, fibroblasts, and vascular endothelial cells that are related to peripheral nerve regeneration. A summary of porous fabrication methods and the resulting pore features are presented in Table 1 and shown in Figure 2.

### 3.1. Templating 

Templating is a traditional and frequently used method for constructing the porous structure of biomaterials, and it allows the fabrication of pores ranging from nanometers to micrometers in size [28]. It is a subtractive method that fabricates porous structures by adding a porogen into a polymer or metal precursor and removing the sacrificial porogen by mechanical stimuli or dissolution such that only the host materials remain. The porosity, size, connectivity, and shape of a porous material can be controlled through careful selection of the porogen morphology [29]. Salt particles and spheres are the most commonly used porogens in biomedical materials. The advantages of the salt template method are presented as follows: (1) biological stability and no chemical reaction with other materials; (2) ease of removal, complete removal by soaking in the aqueous phase; and (3) biocompatibility and no residual biotoxic substances.

Spheres are another commonly used porogen for constructing the 3D porous structures of biomaterials [30]. Compared with the salt leaching method, the shape and size of the spheres can be more strictly controlled; thus, the pore size and 3D morphology of the porous structure of biomaterials can be more precisely regulated. Additionally, a spherical porogen has a higher specific surface area, greater possibility of contact between spheres, and better connectivity of the prepared porous structure. Draghi et al. [31] compared the effects of gelatin microspheres, paraffin microspheres, and NaCl crystals as porogens on the porosity, permeability, morphology, and biocompatibility of scaffolds. They demonstrated that a porous structure fabricated with gelatin and paraffin spheres had higher connectivity and an open pore structure with lower liquid flow resistance through the scaffold.

### 3.2. Freeze-Drying

Freeze-drying or freeze-casting is a common method for preparing porous 3D NGC structures [32]. This technology is based on the vacuum sublimation of ice, freezing water into a solid (phase transition) by cooling and sublimating water molecules in vacuum conditions to obtain porous materials [33]. With the freeze-drying method, the physical 3D morphology of the nerve graft is maintained because the pore structure is formed through the sublimation of the solid ice crystals in a frozen state. Thus, unpredictable deformation of the NGC is avoided [34]. In a previous study, freeze-drying was used to prepare conduits with high porosity. A chitosan conduit with a porosity of 94.24% was prepared by Li et al. [22] using the freeze-drying method. Typically, freeze-drying includes the following steps [35]: (1) freezing; (2) sublimation drying; and (3) secondary drying.

Freezing is a decisive step in constructing porous structures for biomaterials in which the pore size and morphology are tailored by the freezing temperature and method. In the frozen and formed precursor solution, free water that is not bound to the polymer forms ice crystals and sublimates directly from the solid state to the gas state, leaving space to form pore structures. Haugh et al. [33] constructed different pore sizes in collagen/glycosaminoglycan (CG) scaffolds by varying the freezing temperature (−10 °C to −70 °C). The results showed that the pore size decreased with decreasing freezing temperature [36]. Jiang et al. [37] regulated the pore structure of a scaffold by adding different amounts of DMSO to a gelatin solution to control ice crystal formation during the freezing stage. Their results suggested that the formation of ice crystals during freezing could be controlled by adding DMSO, glycerol, or methanol to control the pore size of the scaffold. Because the shape of the pores is consistent with the morphology of ice crystals, the morphology of the pores can be tailored by changing the freezing method. Previous studies [38] have shown that the microchannel structure of the nerve conduit can guide the directional regeneration of axons, and the oriented pore structure, similarly to the microchannel structure, can be prepared via freeze-drying. Wu et al. [39] used rapid unidirectional freezing of liquid nitrogen to control the directionality of ice crystal formation in fabricating a gelatin scaffold with a unidirectional pore structure. The porosity of the scaffold was reduced by increasing the gelatin solution concentration.

### 3.3. Electrospinning Fiber Technology

Electrospinning is a common method for preparing microscale and nanoscale fibers. Electrospun fiber NGCs have unique advantages in peripheral nerve regeneration due to their native extracellular matrix (ECM) mimicking and porous structure that can improve cell–substrate interactions [40]. This technology can result in the production of random or directional fiber films; the cross-space between the fibers forms a porous structure. The size and permeability of the pore structure can be tailored by adjusting the thickness and diameter of the fiber membranes [41]. The porous structure of the electrospun fiber conduit can increase the diffusion of biomolecules and metabolic substances and effectively prevent fibrous scar deposition through pore size adjustment. More important, directional fibers can provide direct physical cues for the directional regeneration of axons from the proximal to the distal end [42].

Aligned fiber nerve conduits have recently been shown to provide mechanical cues for Schwann cell proliferation and axonal outgrowth. An aligned methacrylated silk fibroin electrospun fiber nerve conduit was reported by Chen et al. [43]. Mechanical cues can enhance enriched myelination of Schwann cells through nuclear translocation of Yes-associated protein 1 (YAP) to secrete neurotrophins to support axonal growth. Core–shell electrospun fibers have also been fabricated to endow them with hydrophilicity, early cell adhesion, strong toughness, and degradation properties. As electrospun fiber conduits can provide topological and biological cues to facilitate rapid and directed nerve regeneration, Deng et al. [44] prepared a nanofiber nerve repair graft with an acellular matrix shell and a PCL core. They demonstrated that electrospun fibers with a porous biomimetic extracellular matrix topology could effectively promote nerve fiber maturation and functionalization. Electrospinning has a high specific surface area. Thus, nerve repair drugs can be loaded; the pore structure can promote diffusion of nerve repair drugs or growth factors. Puhl et al. reported an aligned fiber nerve conduit using a pseudouridine-modified neurotrophin-3 (NT-3) mRNA delivery system [45]. The results showed that electrospun fiber conduits could provide ideal structural support and sustained drug-release properties for peripheral nerve regeneration.

### 3.4. Three-Dimensional Bioprinting

Three-dimensional bioprinting is an additive technology that has become a reliable method for constructing biomaterials with complex and precise geometries from 3D model data, especially NGCs with porous structures [46]. The porous structure of NGCs is formed by the spacing between the deposited or solidified material [47]. Although 3D printing technology has been widely used for degradable and nondegradable materials, polymer-degradable materials have been widely used in 3D printing in tissue engineering due to the demand for nerve graft repair materials [48]. 

Compared with traditional pore structure construction methods such as templating and freeze-drying, 3D printing can effectively overcome the limitations of low surface porosity to produce a highly regular and consistent 3D porous structure. According to current research, porous structures prepared using 3D printing technology are macroporous, owing to limited accuracy. However, this method can overcome the problem of reduced mechanical performance of NGCs fabricated using traditional templating, freeze-drying, and electrospinning methods [49]. Qian et al. [50] fabricated multi-layered conduits with macroporous structures by integrating 3D printing and layer-by-layer casting. The results indicated that the 3D porous conduit could improve peripheral nerve regeneration through macroporous structures for the exchange of nutrients and oxygen, with ideal flexibility and rigidity. The construction of biomimetic repair materials similar to natural peripheral nerve tissue in terms of composition and 3D structure has always been the goal of peripheral nerve tissue engineering research. Biomimetic NGCs were fabricated by Vijayavenkataraman et al. [51] using 3D printing technology. A regular square porous structure with a pore diameter of 125 ± 15 μm was fabricated, where the NGCs achieved PC12 cell spreading.

**Table 1 ijms-24-14132-t001:** The representative study of fabrication strategies of porous NGCs and pore structure characteristics.

Fabrication Strategies	Study	Materials	Pore Characteristics	The Assessment of Nerve Regeneration	Limitation
Templating	Li et al., 2020 [52]	Gastrodin and polyurethane	Pore size: 10–60 μm. Open and interconnected.	The proliferation and migration of PC12 cells in vitro.	1. PC12 cells are derived from murine pheochromocytoma and are different from primary neuronal cells; 2. Lacking in vivo animal studies.
	Jeon et al., 2020 [53]	PLGA	Interior to exterior surfaces interconnected.	1. The migration and elongation morphology of PC12 cells; 2. A 10 mm sciatic nerve defect.	The assessment is not imprecise because PC12 cells are not fully representative of primary neuronal cells.
	Fadia et al., 2020 [54]	PCL	Microsphere leaching forms porous structure.	A 5 mm nerve defect in a rhesus macaque model.	Lacking in vitro cytological studies.
	Wang et al., 2015 [55]	PPF-co-PCL	Pore size: 300–400 μm; porosity: 80%.	1. The spreading of PC12 cells; 2. A 5 mm sciatic nerve defect.	The assessment is not imprecise because PC12 cells are not fully representative of primary neuronal cells.
	Cheng et al., 2020 [56]	PVDF/polycaprolactone (PCL)	Pore size: 5.3 μm and 2.7 μm; porosity: 76.2% and 65.5%.	1. The proliferation and migration of Schwann cells; 2. A 15 mm rat sciatic nerve defect.	Lack of evaluation of neuronal cell regrowth.
Freeze-drying	Li et al., 2018 [22]	Chitosan	Pore size: 20–60 μm and 40–100 μm; porosity: 88.19% and 94.24.	A 10 mm rat sciatic nerve defect.	Lacking in vitro cytological studies.
	Ma et al., 2022 [57]	Silk fibroin	Interconnected macroporous structure.	1. Proliferation and differentiation of PC12 cells; 2. A 10 mm rat sciatic nerve defect.	The assessment is not imprecise because PC12 cells are not fully representative of primary neuronal cells.
	Choi et al. 2018 [58]	Decellularized matrix	Pore size: 0.5–20 μm.	A 10 cm rat sciatic nerve defect.	Lacking in vitro cytological studies.
	Shen et al., 2021 [59]	Silk fibroin and PLGA	Interconnected and open.	1. Vascularization of human umbilical vein endothelial cells; 2. A 10 mm rat sciatic nerve defect.	Lacking studies on the effects on glial cells and neuronal cells.
	Li et al., 2020 [60]	Carbon nanotube (CNT)/sericin	Pore size: 12.84 to 346.46 μm; porosity: 85.80%.	1. RSC96 cells were applied to evaluate the biocompatibility; 2. A 10 cm rat sciatic nerve defect.	Lacking in vitro cell experiments.
	Ye et al., 2020 [61]	Gelatin methacrylate (GelMA)	Pore size: 1151 μm, 1564 μm, and 1915 μm.	Supported the survival, proliferation, and migration of PC12 cells.	Lacking in vivo animal studies.
Electrospinning	Chen et al., 2022 [44]	Methacrylated silk fibroin	Directional pore structure	1. Adhesion of Schwann cells; 2. axonal regrowth.	Lacking in vivo animal studies.
	Huang et al., 2017 [62]	PCL	Pore size: 6.5 μm	1. Schwann cell migration; 2. A 15 mm rat sciatic nerve defect.	Lack of proliferation and extension of Schwann cells.
	Zheng et al., 2022 [63]	PCL	Micro–nano morphology	1. Proliferation of Schwann cells; 2. Extension of dorsal root ganglion cells; 3. A 10 cm rat sciatic nerve defect.	Lack of research on the mechanism of vascularization.
	Yoo et al., 2020 [64]	PLCL	Pore size: 2.7 ± 0.6 μm	1. Spreading morphology of PC12 cells; 2. An 8 mm sciatic nerve defect.	The assessment is not imprecise because of the application of PC12 cells in vitro.
	Jaswal et al., 2020 [65]	PCL/gold nanoparticles	The pore size of the spun fiber: 125.85 Å	PC12 cells and Schwann cells were used for biocompatibility and cell morphology studies.	Lacking in vivo animal studies.
3D printing	Namhongsa et al., 2022 [66]	PLCL and PLGA	Pore size: 165 μm and 215 μm	The proliferation and adhesion of Schwann cells.	1. Lacking in vivo animal studies; 2. Lacking in vitro neuronal regeneration studies; 3. Lack of migration and morphology of Schwann cells.
	Qian et al., 2018 [50]	PCL	Pore size: 50 μm	1. Proliferation and attachment of Schwann cells; 2. A 15 mm rat sciatic nerve defect.	Lacking in vitro neuronal cell regeneration studies.
	Vijayavenkataraman et al., 2019 [51]	PCL	Pore size: 125 μm	Proliferation and differentiation of PC12 cells.	1. The assessment is not imprecise because PC12 cells are not fully representative of primary neuronal cells; 2. Lacking in vivo animal studies.

## 4. Implications of Porous NGCs on Peripheral Nerve Regeneration

Peripheral nerve regeneration is a complex physiological process involving Schwann cells, macrophages, fibroblasts, PDGF, FGF, TGF-β, and other cells and the mutual synergistic participation of bioactive factors [67]. The size, permeability, morphology, and interconnection of the pore structure matching the pathophysiological needs of peripheral nerve repair guarantee the synergistic action of these factors [68]. For peripheral nerve repair grafts, pores are typically intended to permit cell migration, maintain an adequate influx of nutrients and oxygen, and eliminate metabolic waste. Thus, the permeability of the pore structure is directly related to its positive role in promoting peripheral nerve regeneration [69]. Pore size and morphology also affect early cellular behaviors such as adhesion, spreading, and migration [70]. Porosity also affects the surface-to-volume ratio, biodegradation rate, stiffness, and mechanical properties of NGCs.

### 4.1. Porous Structure Impact on Physical Properties of NGCs

In clinical applications, ideal NGCs are not evaluated based on a single property; their biodegradation rate, mechanical performance, and permeability should be comprehensively considered. NGC functions are largely responsible for the prognostic efficacy of peripheral nerve regeneration therapies. The 3D porous structure has a direct effect on the properties of NGCs, although the material itself is decisive. It is necessary to fully consider and characterize the effects of porous structures on conduit properties.

#### 4.1.1. Degradability

Degradation of NGCs after in vivo implantation should be coordinated with the process of peripheral nerve regeneration to avoid a lack of structural support for axon regeneration caused by rapid degradation and compression of the newly regenerated nerve caused by slow degradation [6]. At present, the materials for constructing NGCs are mainly chitosan, gelatin, collagen, PLA (polylactic acid), PLGA (polylactic-co-glycolic acid), and high-molecular-weight polymers. After implantation, the polymer NGCs are excreted via chemical degradation and in vivo biodegradation. Chemical degradation refers to the hydrolysis of polymer materials to form small molecular monomers that leave the material body, resulting in a loss of material quality [71]. Biodegradation, also known as bioerosion, refers to the inflammatory response caused by implant stimulation after implantation of polymer materials in the body [72]. The implantation of biomaterials into the body is a process similar to wound healing, but due to the long-term existence of the implanted materials in the body, it will create chronic inflammatory cell infiltration. The duration and extent of this inflammatory reaction mainly depend on the degradation rate and biocompatibility of biomaterials. Polynuclear macrophages caused by human foreign body rejection can alter the microenvironment of the implanted site, such as by a decrease in pH value, thereby accelerating the degradation rate of implant materials [73]. The organism secretes cellular mediators, enzymes, and free radicals to accelerate the hydrolysis of materials, some of which directly cleave their chemical bonds. Methods such as surface modification, porous structure construction, and functional group grafting have been used to improve the immune inflammatory response of medical implantable biomaterials. The long-term immune and inflammatory response will form a fibrous capsule on the surface of the implant, blocking the material signal transmission between the implant and the host, cutting off the nutrition transmission, leading to tissue deformation and patient pain, resulting in implant failure. The pore structure of the NGC can affect the degradation rate of the material through its relationship with the two degradation modes.

Pore size and porosity affect the surface-to-volume ratio of NGCs, as greater porosity and smaller pores increase the specific surface area of the materials. Thus, NGCs with high porosity, small pores, and good connectivity have a larger contact area with biological enzymes and a faster degradation rate [74]. Chemical degradation of polymer materials is mainly related to the degradation activity of the backbone; however, the pore structure can alter the degradation of NGCs by affecting the accumulation of local monomers and pH [75]. Wu et al. [76] studied the in vitro chemical degradation performance of PLGA 3D porous scaffolds based on their porosity and pore size. The results showed that scaffolds with lower porosity or larger pores had thicker pore walls and smaller surface areas, which inhibited the diffusion of acidic degradation products, leading to stronger acid-catalyzed hydrolysis and a faster chemical degradation rate. Song et al. [77] investigated the degradation properties of porous polyester materials using a mathematical modeling strategy. The results indicated that materials with high porosity delayed the degradation process by slowing the autocatalytic reaction. Odelius et al. [78] studied the effects of pore size and porosity on the degradability of PLA scaffolds and degradation product monomers. Their results suggest that larger pore structures can accelerate degradation by autocatalysis.

#### 4.1.2. Mechanical Performance

The mechanical properties of NGCs include axial tensile strength, suture traction strength, radial compressive strength, and bending resistance to evaluate their operability during surgery and the expected protection of newly regenerated axons [79]. NGCs with excellent mechanical performance can effectively avoid suture breakage during operation. More importantly, strong tensile mechanical and anti-distortion properties can reduce the risk of conduit fracture after implantation. When conduits are squeezed by the surrounding tissue, the original round tubular structure is maintained, avoiding compression of the regenerated nerve tissue and blocking nerve regeneration. 

The porosity, pore size, and pore distribution of NGCs affect their mechanical performance. The porous structure reduces the tensile and compressive strengths as the apparent density of the material and its integrity are destroyed [80]. The Young’s modulus of elastomer NGCs, used to evaluate the deformability of elastomers, also changes owing to the pore structure [81]. Wang et al. [82] used the salt template method to prepare porous polyurethane scaffolds with different porosities to investigate the relationship between material porosity and mechanical properties. Their results showed that the mechanical properties of scaffolds were initially positively correlated with the concentration of materials; the mechanical performance of materials with higher porosity decreased at a certain concentration. Chao et al. [83] studied the permeability and mechanical properties of porous scaffolds fabricated using additive manufacturing. A compressive mechanical property test showed that when the porosity was 70%, the elastic modulus and compressive strength of the porous structure tended to decrease with an increase in average pore size. However, when the porosity was 80%, the elastic modulus of the porous structure decreased first and then increased with an increase in average pore size. The compressive strength decreased with an increase in average pore size.

#### 4.1.3. Permeability

Permeability is the most important NGC function provided by the pore structure. Signaling molecule exchange and cell infiltration of conduit walls are the key mediators of NGC success. The permeability of the NGC is determined by the size and connectivity of its pore structure [84]. Conduits with semi-permeable, fully permeable, and asymmetric structures have been prepared; however, the optimal solution remains controversial [2]. The scar barrier caused by fibroblasts invading the NGC is the difference between fully permeable structures, which allow cells, nutrients, and molecular signals to enter the nerve conduit, and semi-permeable structures that do not allow cells to enter the conduit [85]. Vleggeert Lankamp et al. [86] compared the peripheral nerve repair ability of semi-permeable (pore size: 1–10 μm) and fully permeable (pore size: 10–230 μm) conduits. The results showed that the nerve electrophysiology, muscle morphology, and nerve fiber diameter of the semi-permeable nerve conduit group were superior to those of the fully permeable and nonporous structure groups. However, with the discovery that fibroblasts can promote axonal regeneration and Schwann cell proliferation and migration, the communication between cells and axons through paracrine signaling in semi-permeable structures cannot meet the needs of nerve regeneration because peripheral nerve regeneration requires direct contact between cells and newly regenerated axons to regulate the regeneration process [87]. Table 2 summarizes the classification of NGCs according to their passability and their respective characteristics.

Asymmetric high-outflow conduits with large external and small internal pores have become a research focus for promoting the outflow rate of waste products and preventing the invasion of fibroblasts into NGCs. Zhang et al. developed asymmetric NGCs with internal nanoporous structures [88]. With the nanoporous structure inside the tube wall and the asymmetric structure of the hybrid hydrogel on the outer wall, such conduits can spatially limit the distribution of compliance factors, improving the efficacy of peripheral nerve repair. Chang et al. [25] prepared asymmetric PLGA conduits with high outflow using the immersion–precipitation method and implanted them into a 10 mm sciatic nerve defect in rats as an animal model. The silicone conduits of the nonporous structure as the control group showed that the asymmetric NGCs significantly promoted the efficiency of metabolic waste removal and inhibited the invasion of exogenous cells. Microporous PLA conduits with an asymmetric structure were implanted into a rabbit model of a 20 mm sciatic nerve defect in Hus et al. [18]. Eighteen months later, the electrophysiological function recovered to 82%, and the extension of the affected limb recovered to 81%. Tissue sections showed that the newly regenerated nerve fibers formed a normal sciatic nerve bundle.

### 4.2. Porous Structure Determines Bio-Function of NGCs

After peripheral nerve injury, both the distal and proximal ends of the nerve degenerate. Neurons undergo rapid apoptosis if the cell body of the neuron or the axon near the cell body is seriously injured [89]. Peripheral nerve injury can only occur far from the nerve body, or with minor injury, and mainly involves three major processes: Wallerian degeneration, axon regeneration, and terminal nerve reinnervation (Figure 3) [90]. Failure of any of these processes may lead to irreversible neurological dysfunction in patients with peripheral nerve injury. Wallerian degeneration occurs seven days after traumatic peripheral nerve injury, creating a favorable microenvironment for axonal regeneration. Wallerian degeneration is the injury and rupture of the axons of peripheral nerve fibers, blocking the nutritional protection of the axoplasmic flow and the degeneration of the axons and myelin at the distal end of the fibers [91]. The myelin sheath distal to the nerve stumps is swollen and ruptures along with the neuronal fibers; macrophages remove the debris and release growth factors and other substances to stimulate Schwann cell proliferation. Schwann cells migrate to the injured sites and secrete growth factors and adhesion molecules to create a regenerative environment [92]. The proximal end of the injured axon generates a tooth germ that grows at a rate of approximately 1 mm/day along cord-like structures formed by Schwann cells (Bungner bands) to reach the target organ and form new synapses. The axon is remyelinated and the regeneration process is complete [93].

This is only the general physiological process of peripheral nerve regeneration; a more detailed mechanism of regeneration remains to be determined in neuroscience and clinical specialties. The current research directions for peripheral nerve tissue engineering are (1) improving the proliferation and migration of Schwann cells [94]; (2) regulating the macrophage phenotype [95]; (3) inhibiting scar blocking induced by fibroblasts; and (4) promoting angiogenesis of newly regenerated nerve tissue [96]. The pore size, morphology, and porosity can affect the surface roughness, stiffness, and permeability of NGCs, affecting the biological functions of Schwann cells, fibroblasts, macrophages, and vascular endothelial cells (Figure 4).

#### 4.2.1. Schwann Cells

Schwann cells, also known as nerve membrane or sheath cells, are unique glial cells found in the peripheral nervous system. Schwann cells have a variety of important functions [97]: (1) supporting and protecting axons, maintaining a good axon microenvironment; (2) forming a myelin sheath that plays an insulating role on myelinated fibers and accelerates conduction of nerve axons; and (3) having a nutrient-metabolizing effect on nerve axons. After peripheral nerve injury, Schwann cells change from a quiescent state with a stable myelin structure to a proliferative and vegetative state, creating a suitable environment for nerve regeneration [98]. Phenotypic changes of Schwann cells are crucial for Schwann cells to promote nerve regeneration. After Wallerian degeneration at the distal end of the injured nerve, Schwann cells divide, proliferate, participate in the phagocytosis of degenerated axons and myelin debris, and form Bungner bands to guide the growth of regenerated axons. The expression and secretion of nerve growth factors, cytokines, and other active substances can induce and regulate axon regeneration and myelin sheath formation, which are conducive to axon maturation and reinnervation [99].

An appropriate pore structure size can provide attachment points for early adhesion of Schwann cells by changing the surface roughness of the NGC. This is conducive to the early functionalization of cells, such as spreading, and promotes proliferation and migration, ultimately promoting the formation of Bungner bands to guide axon regeneration. Zhang et al. [100] compared Schwann cell proliferation in 3D-printed cold glue/starch composite scaffolds with gaps of 2 mm, 2.5 mm, and 3 mm. After 5 d of culture, there was more significant proliferation of Schwann cells in the 3 mm group than in the other groups. Yu et al. [101] investigated extracellular matrix secretion by Schwann cells on 3D porous chitosan scaffolds and showed that Schwann cells secreted four-fold more laminin and collagen on 3D porous chitosan scaffolds than on the plane. For improvement in the pore size, porosity, and mechanical properties of traditional electrospun scaffolds, an antheraea pernyi silk fibroin scaffold was prepared by Zou et al. [102]. In adjusting the pore size and the distance between the fiber collector and syringe, the viability, penetration, and migration of Schwann cells varied according to the pore size. Pawelec et al. [103] investigated the effect of porous structure on the phenotype of Schwann cells. In vitro cytological studies showed that compared with a smooth glass culture plate, a rough porous structure could increase the expression of Oct-6, MPZ, and MBP and promote differentiation to the mature phenotype.

#### 4.2.2. Macrophages

The traditional view is that the macrophages involved in peripheral nerve injury and regeneration are mainly involved in Wallerian degeneration, engulfment, and clearance of deforming axons and myelin debris [104]. In recent years, with the development and application of cellular and molecular biology techniques, a new understanding of the role of macrophages in peripheral nerve regeneration has emerged. After peripheral nerve injury, macrophages have active phagocytic function and also directly or indirectly participate in peripheral nerve regeneration through their cellular activities and secretion of a variety of growth factors, cytokines, and proteases [99].

In the process of peripheral nerve regeneration, macrophages respond functionally to the stimulation of microenvironment and molecular signals, exhibiting a “classically activated” pro-inflammatory phenotype (M1) and a “selectively activated” anti-inflammatory phenotype (M2) [105]. M1 macrophages are activated by endogenous immune signals induced by injury (e.g., IFN-γ, secreted by Th1 helper T cell subsets). M1 macrophages can produce a large number of oxidative metabolites and pro-inflammatory factors. Therefore, M1 macrophages have the function of promoting inflammatory response [106]. M2 macrophages are mainly activated by cytokines such as IL-4, IL-10, and IL-13 secreted by the helper T cell subset Th2. By producing a large number of anti-inflammatory factors, M2 macrophages promote the formation of new blood vessels, cell proliferation, and the reconstruction of the regenerative microenvironment, thereby promoting the repair and regeneration of damaged tissues [107]. In the process of peripheral nerve regeneration, M1/2 macrophages play different roles, M1 macrophages are mainly involved in Wallerian degeneration in the early stage, and M2 macrophages are mainly involved in the late nerve regeneration process.

Previous studies [108] have shown a correlation between graft surface topography and macrophage polarization. Jiang et al. [37] investigated the effect of pore size and stiffness of scaffolds on macrophage polarization. Macrophages in scaffolds with small pores and softness tended to be activated towards a pro-inflammatory phenotype, whereas cells in scaffolds with large pores and rigidity tended to be activated towards an anti-inflammatory phenotype. Yu et al. [109] investigated the effects of zein microspheres on the induction of immune responses, microsphere size, pore structure, and drug loading during sciatic nerve regeneration. Their results showed that modulation of pore structure could effectively regulate zein-induced immune responses, inhibit neutrophil recruitment, and promote macrophage polarization to the M2 phenotype. High-porosity conduits induce more M2 macrophages to accelerate nerve regeneration. The orientation of the surface topography of biomaterials can promote peripheral nerve regeneration by regulating the polarization of macrophages. Jia et al. [110] found that directional nanofibers increased macrophage elongation along the fiber alignment, induced the development of a pre-healing macrophage phenotype (M2), promoted IL-10 secretion, and inhibited TNF-α factor secretion.

#### 4.2.3. Fibroblasts

In the peripheral nervous system, fibroblasts are the main cellular components of the endoneurium, epineurium, and perineurium (Figure 5a) [111]. There are two sources of fibroblasts after peripheral nerve injury: One is the differentiation and migration of undifferentiated mesenchymal cells and fibroblasts in the adjacent tissue under the action of growth factors and chemoattractants (Figure 5b). Second, growth factors such as PDGF, iFGF, iTGF-β, and iEGF (Figure 5c) stimulate the mitosis and proliferation of fibroblasts in the epineurium and endoneurium of the peripheral nerve at the site of injury (Figure 5d). In the traditional view, scars produced by fibroblasts are detrimental to peripheral nerve regeneration (Figure 5e). Fibroblasts invade the inner part of the NGC to form scar tissue that blocks axonal growth into the target organ. The foreign body reaction caused by graft implantation leads to a long-term inflammatory reaction that forms a fibrous wrapping on the outer surface of the NGC and causes adhesion of the newly regenerated nerve to the surrounding tissue [112]. However, an increasing number of studies have shown that fibroblasts and their secretions play important roles in peripheral nerve regeneration (Figure 5f). He et al. [113] reported that peripheral nerve fibroblasts greatly promoted the outgrowth of motor neuron neurites compared with cardiac fibroblasts. Zhao et al. [87] demonstrated that fibroblast exosomal protein TFAP2C could promote peripheral axon regeneration through the miR-132-5p/CAMKK1 axis pathway. Thus, the traditional concept of blocking fibroblasts from entering the NGC must be reconsidered, and a more advanced pore structure should be explored in future research.

#### 4.2.4. Vascular Endothelial Cells

When a peripheral nerve is injured, the local microenvironment around the injured nerve and the nutrients needed for nerve regeneration are important for repair and reconstruction; these have an important relationship with the local blood supply [114]. Early vascularization of nerve grafts can provide sufficient nutrition for transplanted cells to promote axonal growth and nerve regeneration [115]. The formation of fibrous capsules due to foreign body reactions can be reduced. In addition, vascularization of nerve grafts can gather macrophages from the blood to quickly remove the degeneration products of axons and the myelin sheath after peripheral nerve injury, providing a good channel for new axons to grow to the distal end [116].

The pore structural characteristics of biomaterials such as pore size, porosity, and connectivity significantly affect cell invasion and angiogenesis. Macroporous structures contribute to cell permeability, oxygen and nutrient transport, and the formation of mature vascularized tissues [117]. Some studies have suggested that the effect of pore structure on vascularization is mainly due to the regulation of the immune environment of the material. Yin et al. [118] prepared collagen/chitosan scaffolds with average pore sizes of 160 and 360 μm, respectively, and investigated the effect of pore size on macrophage polarization and its vascularization. The pore size of 360 μm could significantly promote the transformation of macrophages from M1 to M2 and increase the secretion of anti-inflammatory factors (TGF, IL-10, and IGF) and pro-angiogenic factors (VEGF). Studies have shown that collagen–chitosan scaffolds with a larger pore size (360 μm) promote the transformation of macrophages from M1 to M2 and construct a suitable immune microenvironment for vascularization, inducing angiogenesis. Porosity is an important factor in determining blood vessel ingrowth. The porosity of biomaterials is significantly positively correlated with the invasion depth of endothelial cells and the diameter of new blood vessels [119]. Shen et al. [59] used electrospinning to construct a pre-vascularized nerve conduit with high porosity and connectivity. In a 10 mm rat sciatic nerve defect model, pre-vascularized 3D porous NGCs greatly enhanced intra-nerve angiogenesis and promoted peripheral nerve regeneration.

## 5. Conclusions

Significant development of medical tissue-engineered peripheral NGCs has enabled their successful use in clinical practice, which will be more extensive and routine in the future. Many studies have explored the 3D topological morphology of nerve grafting, from relatively closed nerve regeneration chambers to the construction of irregular pore structures, current pore structure instruments with controlled morphology, and asymmetric pore structures. A unified standard for the porous structure of nerve conduits is still lacking, and the ideal porous structure for promoting peripheral nerve regeneration remains controversial. According to recent research results, pore structure characteristics such as porosity, pore size, and connectivity have a comprehensive effect on the nerve regeneration graft, including the degradation rate, mechanical performance, permeability, and other physical properties. The most important factor is that the porous structure can change the roughness, permeability, and hardness of the NGC to have a comprehensive effect on the physiological process of peripheral nerve regeneration.

## Figures and Tables

**Figure 1 ijms-24-14132-f001:**
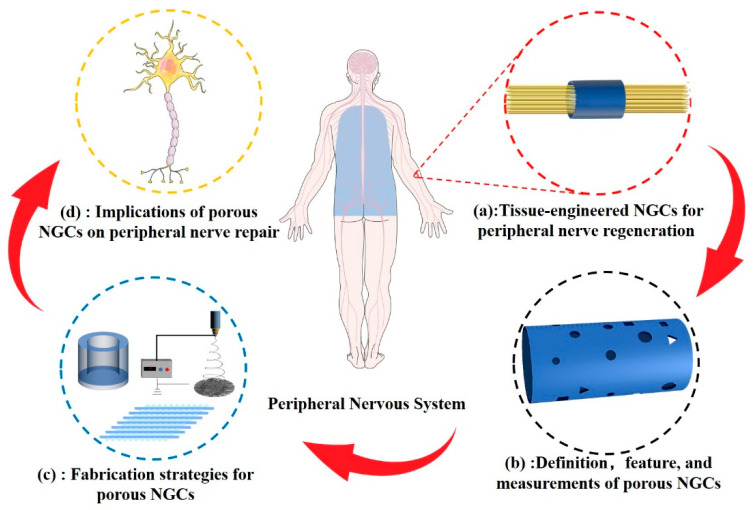
Scope of this review. (**a**) Tissue-engineered NCGs represent a promising treatment strategy that can replace traditional autologous transplantation. (**b**) Definition, feature, and methods for measuring the porous structure of NGCs. (**c**) Strategies for porous NGC fabrication. (**d**) Comprehensive effects of porous NGCs on peripheral nerve regeneration.

**Figure 2 ijms-24-14132-f002:**
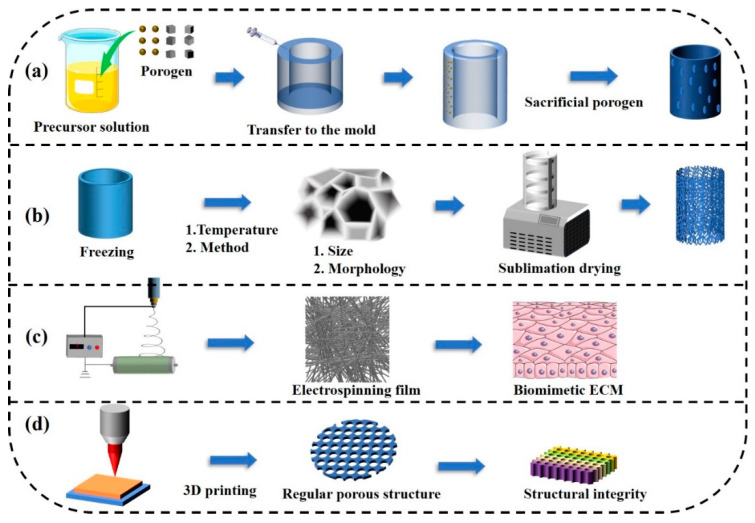
Fabrication strategies for porous NGCs: (**a**) templating; (**b**) freeze-drying; (**c**) electrospinning fiber technology; and (**d**) three-dimensional bioprinting.

**Figure 3 ijms-24-14132-f003:**
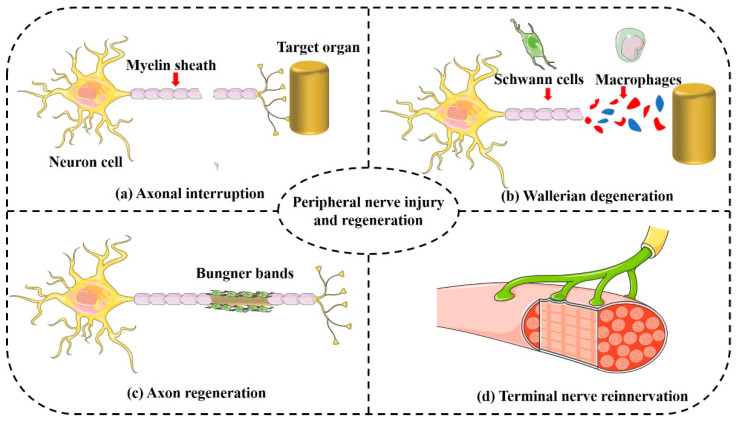
The in vivo pathological process of peripheral nerve regeneration. After peripheral nerve injury (**a**), the damaged axons and myelin sheaths undergo Wallerian degeneration (**b**), the products of which are engulfed by Schwann cells and macrophages. The regenerated axons grow (**c**) into the distal target organ through the Bunger band, completing the reinnervation of the distal target organ (**d**).

**Figure 4 ijms-24-14132-f004:**
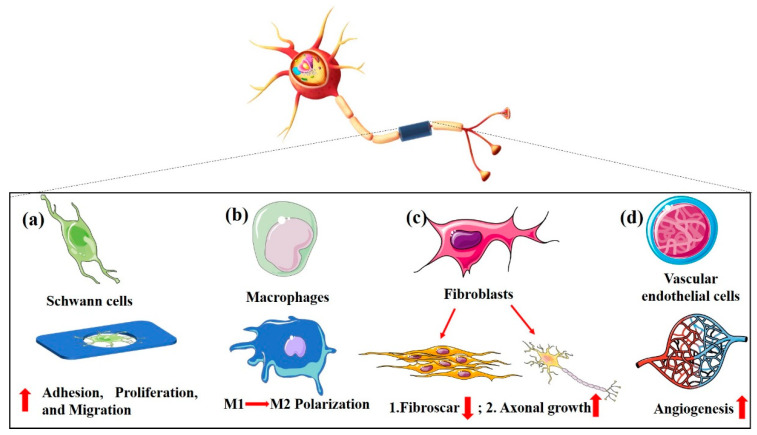
The implications of porous NGCs for peripheral nerve regeneration. (**a**) Pore structure promotes Schwann cell adhesion, proliferation, and migration. (**b**) Pore structure can regulate the polarization of macrophages from pro-inflammatory to anti-inflammatory. (**c**) While fibroblasts promote axonal growth, they also cause fibrous scars. (**d**) The pore structure branch promotes vascularization.

**Figure 5 ijms-24-14132-f005:**
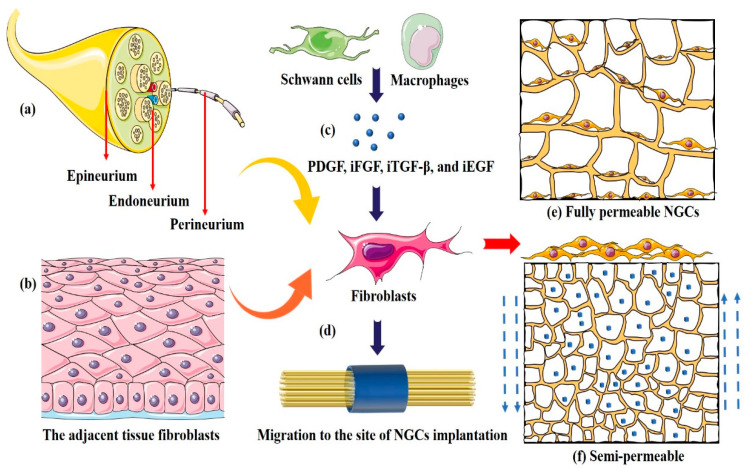
The fibroblasts in the implantation site of NGCs are mainly derived from the endoneurium, epineurium, and perineurium fibroblasts at the injury site (**a**), as well as mesenchymal stem cells and fibroblasts in the adjacent tissues (**b**). They proliferate and differentiate under the stimulation of growth factors secreted by macrophages and Schwann cells (**c**). The fully-permeable conduits (**d**) with large pores (>50 μm) allow cells to pass freely (**e**), while the semi-permeable conduits with small pores (<10 μm) allow only oxygen, nutrients, and intermolecular signaling molecules to pass through (**f**).

**Table 2 ijms-24-14132-t002:** Classification of NGCs according to their passability and their respective characteristics.

Classification	Pore Diameter	Permeability	Advantages in Peripheral Nerve Regeneration	Disadvantageous
Semi-permeable [85]	<10 μm	Nutrients, metabolic waste, and molecular signals.	Avoiding fibrous scars.	Cannot achieve direct signal communication between cells.
Fully permeable [86]	>50 μm	Cells, nutrients, metabolic waste, and molecular signals.	Promote direct signal communication between cells.	Risk of fibrous scar invasion.
Asymmetric structures [88]	External surface pore size > lumen surface pore size.	High outflow.	High effluent efficiency of metabolic waste.	Cannot achieve direct signal communication between cells.

## Data Availability

Not applicable.

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
