# Peer review of "The Porous Structure of Peripheral Nerve Guidance Conduits: Features, Fabrication, and Implications for Peripheral Nerve Regeneration"

_ijms, 2023, doi:10.3390/ijms241814132_

Round 1

Reviewer 1 Report

The review titled “The porous structure of peripheral nerve guidance conduits: features, fabrication and implications for peripheral nerve regeneration” summarized and analyzed the application of porous materials in peripheral nerve repair, including the characteristics of porous structures, construction methods, and biological function implication for peripheral nerve regeneration. The manuscript is clearly presented and well organized. However, I have a few points to make, which I will address in the following:

(1) The distribution of fibroblasts in peripheral nerve tissue is described in Section "4.3.3", but what is the source of fibroblasts invading the nerve conduit from the outside?

(2) Some additions should be made to the studies of the porous structure of nerve conduits summarized in Table 1.

(3)The different ultrastructure of NGC and their application on nerve regeneration should be discussed, like microchannels, aligned fibers etc.

(4) Some of the descriptions in the Figures need more careful editing for clarity.

(5) please check the spelling mistake, e.g. section: Electrospinning fibre technology

the language is easy to follow, but some spelling mistakes should be avoided

Reviewer 2 Report

There are some problems in the manuscript:

1.It is not a comprehensive review, with only 3 figures

2.Lack of Originality.

3.There are too little valuable information.

4. Lack of author's own opinion, just a review/repetition of what has been published

5. Lack of future trend

6. Lack of summative Table and Data.

7. Lack of discussion and comparsion of reviwed data.

Some errors should be corrected.

Reviewer 3 Report

This paper is well written and have consolidated the in-depth scientific understanding on the effect of peripheral nerve guidance on nerve regeneration. Paper can be accepted after minor revision. To improve the quality of the manuscript the review comments are appended below.

1)     The effect of pore diameter on nerve regeneration was well discussed. However, it is good to discuss the different types of scaffold pore which could be in the form of gyroid or elliptical or spherical structure that determining the nerve regeneration. I encourage you to have a look at this paper.

https://doi.org/10.1016/j.matpr.2022.05.469

2)     To interest wider audience, it is good to include the degradation mechanism of the peripheral nerve guide in normal and inflammatory conditions. During the normal condition the degradation would be much slower. However, during high oxidative stressed condition the pH ~3- 3.5 where the degradation would be much faster. To have some idea on degradation mechanism I encourage you to have a look at this following paper.

 https://doi.org/10.1016/j.jallcom.2023.169852

Reviewer 4 Report

The authors present a review on the porous structure of nerve guidance conduits and their implications in the treatment of long-segment peripheral nerve injuries. The authors should be commended for for summarizing such a wide body of literature so comprehensively. This article will serve as a roadmap for researchers in this space for the coming years.

Author Response

Thank you for your comments on this work